# Functionalization of 3D-Printed Titanium Scaffolds with Elastin-like Recombinamers to Improve Cell Colonization and Osteoinduction

**DOI:** 10.3390/pharmaceutics15030872

**Published:** 2023-03-08

**Authors:** Jordi Guillem-Marti, Elia Vidal, Alessandra Girotti, Aina Heras-Parets, Diego Torres, Francisco Javier Arias, Maria-Pau Ginebra, Jose Carlos Rodriguez-Cabello, Jose Maria Manero

**Affiliations:** 1Biomaterials, Biomechanics and Tissue Engineering (BBT), Department of Materials Science and Engineering (CEM), Universitat Politècnica de Catalunya—BarcelonaTech (UPC), 08019 Barcelona, Spain; 2Barcelona Research Center in Multiscale Science and Engineering, Universitat Politècnica de Catalunya—BarcelonaTech (UPC), 08019 Barcelona, Spain; 3Smart Devices for NanoMedicine Group, Department of Biochemistry and Molecular Biology and Physiology, University of Valladolid, 47011 Valladolid, Spain; 4Unidad de Excelencia Instituto de Biomedicina y Genetica Molecular (IBGM), Universidad de Valladolid and Consejo Superior de Investigaciones Cientificas (CSIC), 47003 Valladolid, Spain; 5Institute for Bioengineering of Catalonia (IBEC), Barcelona Institute of Science and Technology (BIST), 08028 Barcelona, Spain; 6BIOFORGE (Group for Advanced Materials and Nanobiotechnology), Centro de Investigación Biomédica en Red—Bioingeniería, Biomedicina y Nanomedicina (CIBER-BBN), University of Valladolid, 47011 Valladolid, Spain

**Keywords:** 3D printing, elastin-like recombinamers, functionalization, osseointegration, titanium

## Abstract

The 3D printing of titanium (Ti) offers countless possibilities for the development of personalized implants with suitable mechanical properties for different medical applications. However, the poor bioactivity of Ti is still a challenge that needs to be addressed to promote scaffold osseointegration. The aim of the present study was to functionalize Ti scaffolds with genetically modified elastin-like recombinamers (ELRs), synthetic polymeric proteins containing the elastin epitopes responsible for their mechanical properties and for promoting mesenchymal stem cell (MSC) recruitment, proliferation, and differentiation to ultimately increase scaffold osseointegration. To this end, ELRs containing specific cell-adhesive (RGD) and/or osteoinductive (SN_A_15) moieties were covalently attached to Ti scaffolds. Cell adhesion, proliferation, and colonization were enhanced on those scaffolds functionalized with RGD-ELR, while differentiation was promoted on those with SN_A_15-ELR. The combination of both RGD and SN_A_15 into the same ELR stimulated cell adhesion, proliferation, and differentiation, although at lower levels than those for every single moiety. These results suggest that biofunctionalization with SN_A_15-ELRs could modulate the cellular response to improve the osseointegration of Ti implants. Further investigation on the amount and distribution of RGD and SN_A_15 moieties in ELRs could improve cell adhesion, proliferation, and differentiation compared to the present study.

## 1. Introduction

In recent decades, titanium (Ti) and its alloys have been widely used for bone implant purposes due to their proper mechanical properties, good biocompatibility, and high resistance to corrosion [1,2,3]. Recently, the incorporation of Ti in the growing 3D printing industry has enabled the design of 3D-printed Ti scaffolds suitable for different bone/orthopedic applications [4]. The design of 3D-printed Ti structures with customized shapes and geometries, together with the possibility of controlling its porosity and pore size, has revolutionized the biomedical field by offering the possibility of tuning mechanical properties and improving the biological response simultaneously [5].

In this regard, 3D-printed structures can be colonized by cells to promote bone tissue ingrowth within the scaffold to obtain better tissue integration and mechanical stability. However, the osseointegration and osteoconduction capacities of Ti are limited due to its lack of bioactivity [6]. To overcome this limitation, different strategies have been adopted in recent decades, mostly grouped into physical or chemical surface modifications. On the one hand, physical surface modifications such as the alteration of nano- and/or micro-roughness have produced significant advances in the osseointegration of Ti implants, although they are difficult to apply to 3D-printed scaffolds. On the other hand, chemical surface modifications, focused on tethering biomolecules to provide adhesive and/or functional moieties at the implant surface [7,8,9], are more applicable for 3D-printed scaffolds. Immobilization of these bioactive molecules on the surface may provide the cells with signals for adhesion, differentiation, and, in the case of porous scaffolds, cellular migration into the structure.

Several proteins from the extracellular matrix (ECM) have been used to accomplish these functions, such as collagen [10], fibrinogen [11], and fibronectin [12]. Although they demonstrate improved osseointegration capacities when immobilized onto biomaterials, natural proteins are difficult to obtain or produce, which hampers their clinical translation [13]. In contrast, the use of synthetic recombinant protein fragments [14,15] or small peptides [16] is a promising alternative since huge yields can be obtained, thus considerably reducing the costs.

In this context, the use of genetic engineering methodologies enables the design of highly complex functional proteins or polymeric proteins with several properties [17], such as elastin-like recombinamers (ELRs), which are synthetic proteins whose sequences are derived from certain peptides of natural elastin [18,19]. ELRs retain the elastin mechanical properties, thermo-responsiveness, non-immunogenicity, and excellent biocompatibility and present a strong tendency to self-assemble depending on the environmental conditions [20]. Their sequence is based on the repetitions of the conserved pentapeptide Val-Pro-Gly-X-Gly (VPGXG), where the guest X position can accommodate any natural amino acid but proline. Interestingly, ELRs can be genetically engineered to include additional amino acids or domains that tune their bioactivity without affecting the elastin properties. For instance, the addition of RGDS or REDV, small adhesive peptides found in different ECM proteins, into ELRs has been demonstrated to increase cell adhesion and colonization when immobilized onto different materials [21,22,23]. However, additional signals are required to induce the differentiation of bone cells and, ultimately, bone regeneration. The use of the small peptide SN_A_15 (DDDEEKFLRRIGRF) derived from the N-terminal 15-amino acid residue of statherin, for instance, has shown similar mineralization capacities to the native protein when inserted into ELRs [24,25]. Statherin is a salivary protein that plays a key role in the nucleation and growth of hydroxyapatite in the oral cavity by binding calcium ions and hydroxyapatite on its N-terminal position [26]. However, the SN_A_15 peptide does not promote cell adhesion, hence requiring the presence of additional signals. Noteworthy, in all cases, ELRs containing cell-adhesive and/or osteoinductive motifs were immobilized onto 2D materials without considering the possible influence of 3D features.

The main objective of the present study was to Improve the osseointegration capacities of 3D-printed Ti scaffolds by functionalizing their surfaces with genetically modified ELRs. The novelty of this work relies on the fact that, to our knowledge, ELRs or genetically modified ELRs have not been used before in 3D structures. To this end, we covalently attached ELRs that were genetically engineered to incorporate in their sequence the following biomolecules: (i) a dodecapeptide peptide derived from fibronectin that included RGD to promote cell adhesion; (ii) an SN_A_15 peptide from statherin to mediate hydroxyapatite nucleation; or (iii) both RGD and SN_A_15 sequences together for a synergistic effect. Non-modified ELRs and a 3D-printed Ti scaffold with no ELRs were used as a control. The main goal of the present study was to assess the effect of genetically modified ELRs functionalized onto 3D-printed Ti scaffolds in mesenchymal stem cell colonization, differentiation, and mineralization.

## 2. Materials and Methods

### 2.1. Printing of Scaffolds

Cylindrical shape scaffolds with cubic internal pores obtained by a 0-90º pattern, where the filaments in each level are placed perpendicularly to the level beneath them, were selected among other possible geometries (Figure 1). This geometrical design allows for a more open structure with higher porosities compared to other common structures (e.g., honeycomb), although sacrificing to some extent their mechanical behavior. In addition, this design allows for highly interconnected pores whose dimensions can be easily controlled by the strut dimensions and the distance between the filaments.

The 3D Ti scaffolds were fabricated as previously described [27]. Briefly, Ti powders (Alfa Aesar, Haverhill, MA, USA) were mixed with 30% Pluronic F-127 (Sigma-Aldrich, St. Louis, MO, USA) in water using a SpeedMixer DAC 150.1 FVZ (FlackTek Inc., USA) asymmetric centrifugal mixer system at 3500 rpm for 5 min (Figure 1). The obtained ink was extruded through a 410 μm diameter nozzle syringe (Smooth Flow Tapered Tips, Nordson EFD, Westlake, OH, USA) using a DIW device (BCN3D printer + Paste cAster, Fundació CIM, Barcelona, Spain). The cubic geometry of the internal pores was designed using the SolidWorks software (SolidWorks Corp., Waltham, MA, USA), and the printing parameters were adjusted using the Slic3r free software (http://slic3r.org/, Accessed on 24 May 2022). The nozzle diameter defines the diameter of the printed strut. The layer height was set to 350 μm, the infill density was 45%, and the infill speed was 15 mm/s. The infill density is related to the density of the 3D-printed scaffold. Increasing this value implies a higher density of the scaffold. An infill of 45% was set as the optimal balance between the low-density structure and the achievement of an architecture capable of supporting its own weight. Using these parameters, scaffolds of 75% porosity were obtained after sintering with pores in the range of 316–332 um [27], which are in the optimal range for cell colonization [28] and osteointegration [29]. The samples were designed to obtain 13 mm diameter and 5 mm height scaffolds containing a total of 12 alternated filament layers.

After printing, the Pluronic F-127 was removed by a thermal treatment at 280 °C in an oven. Subsequently, the scaffolds were sintered in a tubular furnace (Carbolite, Hope Valley, UK) under high-vacuum conditions (10^−5^ mbar). The temperature was raised at a constant rate of 2.5 °C min^−1^ until 1350 °C, sintering the scaffolds for 3 h.

### 2.2. Synthesis and Characterization of Elastin-like Recombinamers

The ELR genes were synthesized following recombinant DNA technologies, and the protein recombinamers were produced using the *Escherichia coli* expression system as previously described [30,31]. The ELRs were purified from the bacterial biomass by several cycles of temperature-dependent reversible precipitation [32]. Finally, the ELRs were dissolved and dialyzed against ultrapure water at 4 °C, sterilized by filtration, and freeze-dried. The ELRs’ purity grade was determined by dodecyl sulfate polyacrylamide gel electrophoresis (SDS-PAGE), nuclear magnetic resonance spectroscopy (NMR), and Fourier-transform infrared spectroscopy (FT-IR), as described elsewhere [33].

### 2.3. Scaffold Functionalization

Figure 2 shows a schematic diagram of the process followed to functionalize the scaffolds. The scaffolds were cleaned and activated by an oxygen plasma treatment for 5 min at 12 MHz in a Femto low-pressure plasma equipment (Diener Electronic, Ebhausen, Germany). Samples were then immersed in 0.5 M (3-chloropropyl)triethoxysilane (CPTES) and 0.05 M diisopropylethylamine (DIEA) in anhydrous toluene for 1 h at 70 °C under agitation in a nitrogen atmosphere to avoid the incorporation of water in the solution. The samples were ultrasonically cleaned with anhydrous toluene; washed with toluene, ethanol, isopropanol, distilled water, and acetone; and finally dried with nitrogen. Afterward, the samples were incubated overnight in a solution of 0.05 mg/mL ELR containing 0.5 mM Na_2_CO_3_. After incubation, the samples were rinsed thrice in a phosphate-buffered saline (PBS) and blocked with 1 w/v% bovine serum albumin (BSA) in PBS for 30 min. Before cellular assays, the samples were sterilized in 70% ethanol for 30 min and washed with PBS three times. Non-functionalized Ti scaffolds were used as the control.

### 2.4. Characterization of ELR Functionalization

The presence of ELRs functionalized onto the Ti scaffolds was inspected by fluorescence visualization. To this end, the HRGD was fluorescently labeled with FITC in a molar ratio of 3:1 FITC to ELR and using an amine-reactive derivative of fluorescein dye (NHS-Fluorescein, Thermo Fisher Scientific, Waltham, MA, USA). Briefly, HRGD and FITC were dissolved in a DMF-DMSO mixture (1:1 *v/v*) at room temperature for 6 h. The FITC solution was slowly poured into the ELR solution and incubated for 24 h. Afterward, the reaction was precipitated in diethyl ether (Honeywell, Wabash, IN, USA), washed twice with acetone (Scharlab, Sentmenat, Spain), and allowed to dry under a vacuum at room temperature. The ELRs were dissolved in water at 4 °C and dialyzed against ultrapure water, sterilized by filtration, and freeze-dried. After functionalization, the scaffolds were immersed in liquid nitrogen and longitudinally crushed. The distribution of fluorescently labeled HRGD was assessed by visualization of the scaffolds using an MVX10 Research Macro Zoom Microscope (Olympus, Tokyo, Japan).

### 2.5. Cell Culture and Maintenance

Rat mesenchymal stem cells (rMSCs) were isolated from the tibias and femurs of young Lewis rats and expanded in Advanced DMEM medium supplemented with 10% fetal bovine serum (FBS), 2 mM L-glutamine, penicillin/streptomycin (50 U ml^−1^ and 50 µg ml^−1^, respectively), and 20 mM HEPES buffer solution, all from Thermo Fisher Scientific (USA). The cells were maintained at 37 °C in a humidified atmosphere and 5% CO_2_. In all the experiments, the cells from passage 4 were used and seeded at a density of 50 × 10^3^ cells/sample in a serum-free medium.

### 2.6. Cell Adhesion and Proliferation

Cells were quantified after 4 h of adhesion or allowed to proliferate for 7, 14, 21, and 28 days after replacing the medium with a complete medium containing 10% FBS. No osteogenic medium was used. After each incubation period, the medium was removed, and the cells were washed and lysed using 300 µL of Mammalian Protein Extraction Reagent (M-PER; Thermo Fisher Scientific). The number of cells at each specified time was quantified using the Cytotoxicity Detection Kit (LDH) (Roche Applied Science, Penzberg, Germany) following the manufacturer’s instructions. The measurements were obtained spectrophotometrically at 492 nm using a Synergy HTX multi-mode microplate reader (Bio-Tek Instruments, Winooski, VT, USA), and the cell numbers were extrapolated using a calibration curve with decreasing numbers of cells. The tolerance for the peak absorbance is ±3 nm while the tolerance for the optical density alignment is ±0.015 AU in this equipment. The error in the last test performed was ±0.007 AU in the optical density measurement, and the repeatability of results showed a deviation of ±0.001 AU.

### 2.7. Cell Colonization and Morphology

After 4 h or 7, 14, 21, and 28 days of incubation, the cells were fixed with 2.5% glutaraldehyde solution in phosphate buffer (PB) for 1 h at 4 °C. The samples were subsequently immersed in osmium tetroxide solution for 1 h and rinsed with distilled water. Then, the scaffolds were dehydrated by immersion in an ethanol series (50%, 70%, 90%, and 96%) and completely dehydrated in hexamethyldisilazane for 15 min. Dried samples were cut horizontally down the mid-section after immersion in liquid nitrogen, and cell penetration into the scaffolds was visualized by Zeiss Neon 40 scanning electron microscopy (SEM; Carl Zeiss, Jena, Germany) using 5 kV voltage. The percentage of the area occupied by cells on filaments four and five (from the top, where cells were seeded downwards) was quantified in each condition using the ImageJ software (National Institute of Health, Bethesda, MD, USA).

### 2.8. Cell Differentiation and Mineralization

Cell differentiation into the osteoblastic lineage was evaluated by measuring the alkaline phosphatase (ALP) activity as a differentiation marker. For this purpose, the same extracts obtained in the cell proliferation assay were used for quantifying the ALP activity with a SensoLyte pNPP Alkaline Phosphatase Assay Kit (AnaSpec Inc., Fremont, CA, USA). Enzymatic activity was evaluated spectrophotometrically at 405 nm for each specified time point using a Synergy HTX multi-mode microplate reader (Bio-Tek Instruments, USA), and the results were normalized versus their corresponding cell number obtained in the proliferation assay. Mineralization was assessed after culturing the cells for 28 days on scaffolds. Then, the cells were fixed in 4% paraformaldehyde solution in PBS for 15 min and rinsed with distilled water. Calcium deposits were stained with 500 µL/scaffold of 40 mM Alizarin Red S (Sigma-Aldrich, USA) for 20 min by gentle shaking. Excess dye was removed by several washings with distilled water, and the staining was extracted by incubation in 10% cetylpyridinium chloride with 10 mM NaH_2_PO_4_. Supernatants were measured spectrophotometrically at 570 nm using a Synergy HTX multi-mode microplate reader (Bio-Tek Instruments, USA), and the results were normalized versus their corresponding cell number obtained in the proliferation assay.

### 2.9. Statistical Analysis

Data are presented as mean values ± standard error of the mean. Experiments were performed in triplicate using three replicates per group. Statistically significant differences between the groups (*p*-value < 0.05) were analyzed by the Kruskal–Wallis non-parametric test followed by the Mann–Whitney U test with Bonferroni correction using SPSS statistics software (IBM, Armonk, NY, USA).

## 3. Results and Discussion

### 3.1. ELR Synthesis and Characterization

The recombinamers studied in this work were based on the repetitions of the elastin pentapeptide containing isoleucine (I) or lysine (K) as guest amino acids, VPGIG, and VPGKG. The lysine-free γ-amino groups were used for functionalization through their interaction with CPTES. This basic sequence involved the ELR as a control, coded as IK. Moreover, modified versions of this basic ELR were synthesized, containing different bioactive moieties: (i) a peptide derived from statherin (SN_A_15), which is involved in the nucleation of hydroxyapatite (H3); (ii) the RGD peptide of human fibronectin involved in cellular adhesion (HRGD); and (iii) the combination of both bioactive peptides into one ELR (H4R4). The different ELRs’ compositions and molecular weights (MWs) are shown in Table 1.

The purity of the different ELRs determined by SDS-PAGE was greater than 95% (Figure 3A). However, the ELRs containing the statherin domain presented a partial degradation. All ELRs showed electrophoretic mobility delay with respect to the theoretical molecular weight. However, the migration of the ELRs was consistent with the expected MW differences with respect to each other. This phenomenon is due to the high content of hydrophobic amino acids of the synthetic protein polymers which results in a slow migration of the ELRs and an apparently higher molecular weight, as previously observed by other authors [34,35].

The degree of purity was also corroborated by the acquisition of the infrared spectra (Figure 3B). The correspondence of the ELRs’ profiles was verified, and undesired contaminants were not detected. In addition, the proton NMR spectra (Figure 4) confirmed the ELRs’ purity and identity by the determination of the relative amount (ratio) and absolute content of amino acids [36].

### 3.2. Scaffolds Functionalization

In previous works, we demonstrated that the use of silanes, specifically 3-chloropropyltriethoxysilane (CPTES) or 3-aminopropyltriethoxysilane (APTES), enables the covalent immobilization of ECM proteins on plasma-activated Ti surfaces [10,14,15,37]. Herein, we verified that the same strategy could be applied to 3D-printed Ti scaffolds since, after crushing the sample, we observed fluorescently labeled HRGD distributed throughout the scaffold (Figure 5A). In high-magnification images, we observed that the HRGD was homogenously distributed along the Ti filaments (Figure 5B). Since the linking mechanism between HRGD and CPTES silane is through the lysine in the elastin-like repetitions of the former, we assumed that all the different studied ELRs were also homogeneously distributed throughout the Ti scaffolds.

### 3.3. Cell Adhesion and Proliferation

The number of rMSCS adhered to the scaffolds was higher on HRGD-functionalized surfaces than the other functionalized scaffolds and the non-functionalized scaffolds (Figure 6A). This enhancement in cell adhesion could be associated with the presence of the RGD motif in the ELR sequence, which mediates cell adhesion through integrin interaction [38]. However, in previous studies, it was observed that the addition of RGD in ELRs did not increase the number of adhered cells compared to ELRs without the RGD sequence [39,40,41], and this was interpreted as an indication that the cells were able to selectively recognize and adhere to elastin-derived peptides, irrespective of the presence of RGD [42]. Therefore, we hypothesize that there might be a synergistic effect between HRGD and the roughness of the scaffold filaments, which was approximately 8 μm [37]. In contrast, the presence of the statherin motif in the RGD-containing ELR, i.e., H4R4, decreased the adhesion of cells compared to HRGD-functionalized scaffolds. This might be associated with the length of the H4R4 polymer, which implies a reduction in the amount of RGD sequences in the same area compared to the HRGD ELR.

In addition, HRGD enhanced not only cell adhesion but also the proliferation of rMSCs cultured on functionalized Ti scaffolds (Figure 6B). In previous studies, we demonstrated that the presence of RGD in ELR stimulated the proliferation compared to the non-functionalized samples or ELR-functionalized surfaces, i.e., IK [40,43]. The cells proliferated adequately on all the other surfaces, although lower levels were observed for the statherin-containing ELR-functionalized scaffolds, i.e., H3-functionalized Ti scaffolds.

### 3.4. Cell Colonization

To assess the penetration of cells throughout the Ti scaffolds, the cells were incubated at different times, and scaffold cross-sections were observed by SEM. Cells adhered mostly in the upper filaments of the functionalized and non-functionalized Ti scaffolds after 4 h of incubation (Appendix A). It is noteworthy that more cells were observed on RGD-containing ELRs-(HRGD and H4R4) functionalized Ti scaffolds compared to the other conditions (Figure 7). It has been previously observed that the presence of RGD in ELRs stimulates the adhesion of cells more rapidly compared to non-RGD ELRs [44,45].

The cells were able to migrate through the entire Ti scaffolds from the upper to the lower part over the incubation periods (Appendix A), as the cells were observed even on the surface of the bottom filaments at 28 days of culture (Appendix A). The scaffold pore size, approximately 400 μm [37], allowed the successful penetration of the cells in the present study. It has been previously described that pores from 300 to 600 µm in diameter are required to allow the migration of cells into porous structures in in vitro studies [28]. Smaller pore sizes produce the accumulation of cells in external parts of scaffolds that are detrimental not only to cell migration but also to oxygen and nutrient supply [46]. Of note, the number of cells on the filaments in the lower part of the scaffolds (parallel filaments 4 and 5, from the top downwards) was larger for the ELR-functionalized compared to the non-functionalized Ti scaffolds, especially with long incubation periods (Figure 8). Notably, there were few differences between filaments 4 and 5 within each condition, demonstrating that the cells could colonize the whole scaffold.

### 3.5. Cell Differentiation and Mineralization

The results of the ALP activity indicated that all the surfaces induced differentiation of rMSCs after 7 days, since the values obtained were significantly higher than those obtained after 4 h of culture, where no ALP activity was detected (Figure 9A). It is important to mention that a non-osteogenic medium was used in all the conditions. Remarkably, the differentiation of rMSCs was found even in non-functionalized Ti scaffolds. This could be attributed to two reasons. On the one side, it has been reported that geometrical effects associated with the macro-structure of the 3D porous constructs may play a role in cell differentiation [47]. On another side, the combined micro-/nano-roughness generated by the 3D printing methodology itself and the sintering process [37] may also stimulate rMSCs differentiation [48].

Noteworthily, the functionalization of scaffolds with the ELR containing the SN_A_15 peptide fragment from statherin (H3) responsible for the nucleation of hydroxyapatite produced an increase in ALP activity at 14 days compared to the other conditions (Figure 9A). This high ALP activity correlated with the lower proliferation of rMSCs on the H3-functionalized Ti scaffolds mentioned above. This is because precursor cells continuously divide until they acquire a fully differentiated state, and terminal differentiation coincides with proliferation arrest [49]. Although the exact process by which the bioactive peptide SN_A_15, present on H3, stimulates osteoblastic differentiation is not fully understood, it was previously speculated that the negative charges present at the N-terminal site of the SN_A_15 peptide might mediate the binding of calcium phosphate ions [50]. It has been observed that the accumulation of calcium phosphate ions induced by the statherin peptide sequence has an osteoinductive effect on the MSCs cultured on H3 membranes [40]. In fact, the cells grown on H3 scaffolds exhibited higher levels of mineralization in the present study (Figure 9B). It is important to mention that the ELR containing both RGD and SN_A_15 (H4R4) did not stimulate differentiation and mineralization to the same extent as the ELR containing only SN_A_15 (H3). This particular behavior has been previously observed and is attributed to the fact that the number of SN_A_15 epitopes in H3 is higher than the number exhibited in H4R4 for the same polymer weight [39,40]. Although mineralization was lower in H4R4 than in H3 scaffolds, it was higher compared to HRGD and bare Ti (Figure 9B).

One of the main limitations of the current study is the use of one specific 3D structure without considering other scaffold topologies. It is well known that several parameters including the pore size and geometry [46], or the scaffold architecture [51], have a significant effect on the cellular response and on osseointegration of the scaffold. As a proof of concept, in the present study we functionalized the scaffolds of cubic internal pores because it allows for a more open structure with higher porosities. Further experiments will demonstrate that the method may be applied to other 3D structures. In addition, another limitation is that the current work was performed in static conditions, while it is well known that dynamic cultures strongly influence the behavior of cells [51]. The influence of dynamic culturing on the MSCs’ response on ELR-functionalized scaffolds will be evaluated in further studies.

## 4. Conclusions

The functionalization of 3D-printed Ti with ELRs has been demonstrated to be a promising strategy for tuning the interaction of MSCs with the filaments and colonization of the scaffold. The modification of ELRs with adhesive and/or osteoinductive motifs has been demonstrated to trigger different cell behaviors. The addition of RGD in the ELR sequence stimulated the adhesion and spreading of cells and their migration and colonization of scaffolds. In contrast, the presence of the hydroxyapatite nucleation fragment of statherin promoted MSC differentiation into the osteoblastic lineage, even in the absence of an osteogenic cell culture medium, although it decreased cell adhesion and proliferation. Although the combination of both adhesive and osteoinductive motifs in the ELR improved adhesion and proliferation compared to the statherin-containing ELR, the osteoinductive potential was lower. Thus, the functionalization of 3D Ti scaffolds with this statherin-containing ELR could be of interest in improving the osseointegration capacities of Ti scaffolds. Further research exploring the in vivo behavior of the ELR-functionalized Ti scaffolds will shed light on their osseointegration potential.

## Figures and Tables

**Figure 1 pharmaceutics-15-00872-f001:**
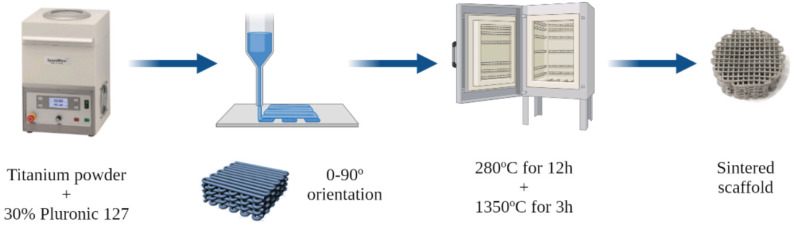
Schematic diagram of the 3D printing process of Ti scaffolds.

**Figure 2 pharmaceutics-15-00872-f002:**
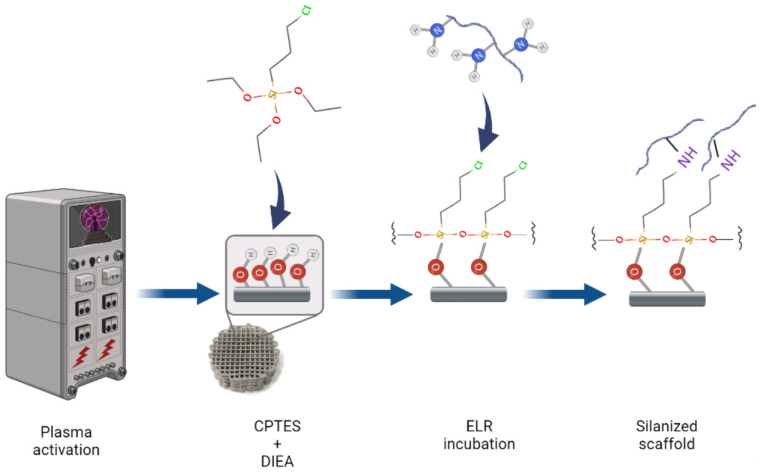
Schematic diagram of the functionalization process.

**Figure 3 pharmaceutics-15-00872-f003:**
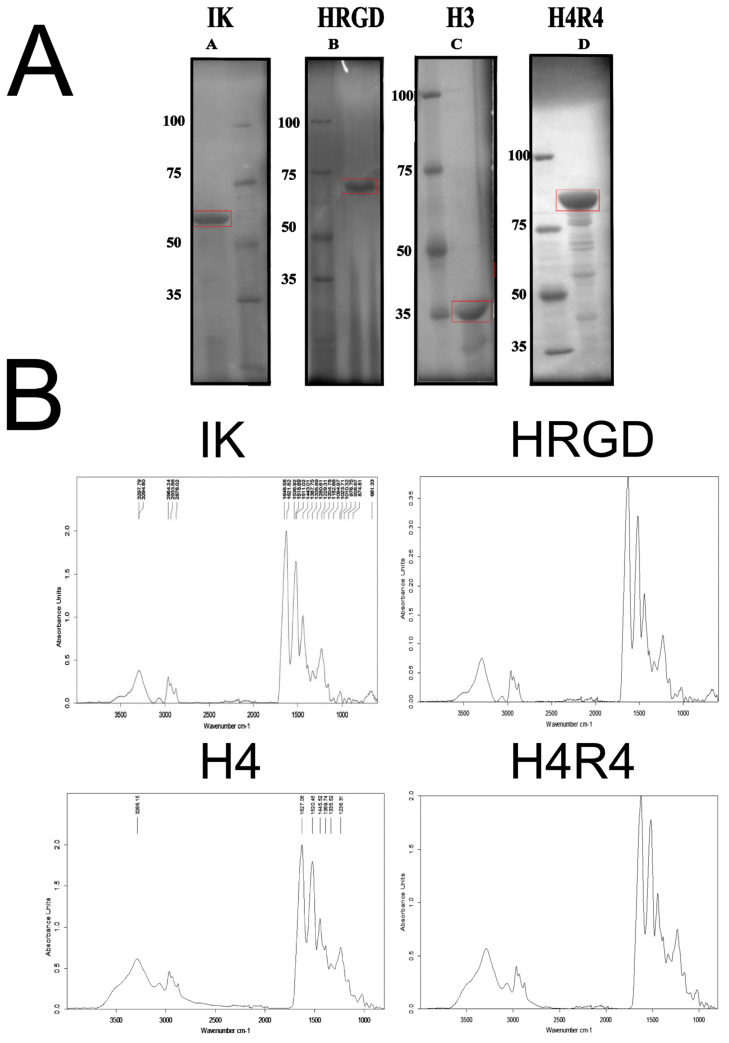
Characterization of the purity of ELRs. (**A**) SDS-PAGE of purified ELRs. The MW of the SDS-PAGE protein marker bands is indicated on the left side. The bands corresponding to the purified ELR are included in a red box. The theoretical MW for IK, HRGD, H3, and H4R4 are 52 kDa, 60.7 kDa, 31.9 kDa, and 80.7 kDa, respectively. (**B**) FTIR spectra of the purified ELRs.

**Figure 4 pharmaceutics-15-00872-f004:**
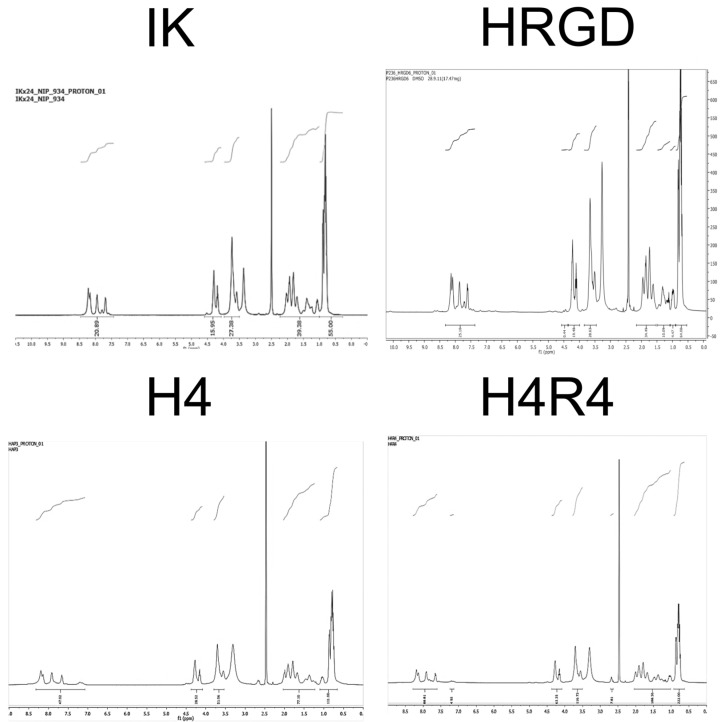
Proton NMR spectra of the purified ELRs.

**Figure 5 pharmaceutics-15-00872-f005:**
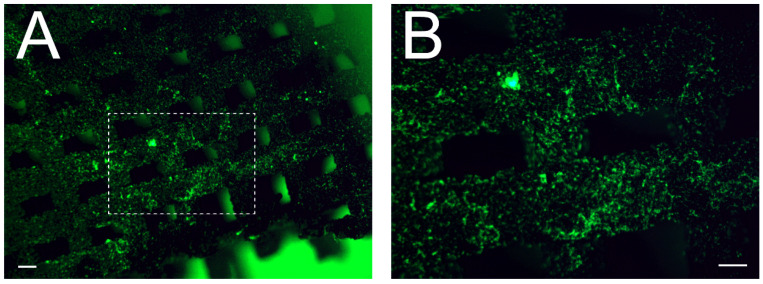
Characterization of scaffold functionalization. Low- (**A**) and high-magnification (**B**) fluorescence images of HRGD-functionalized Ti scaffolds. Samples were crushed and longitudinally observed. Scale bar denotes 200 μm.

**Figure 6 pharmaceutics-15-00872-f006:**
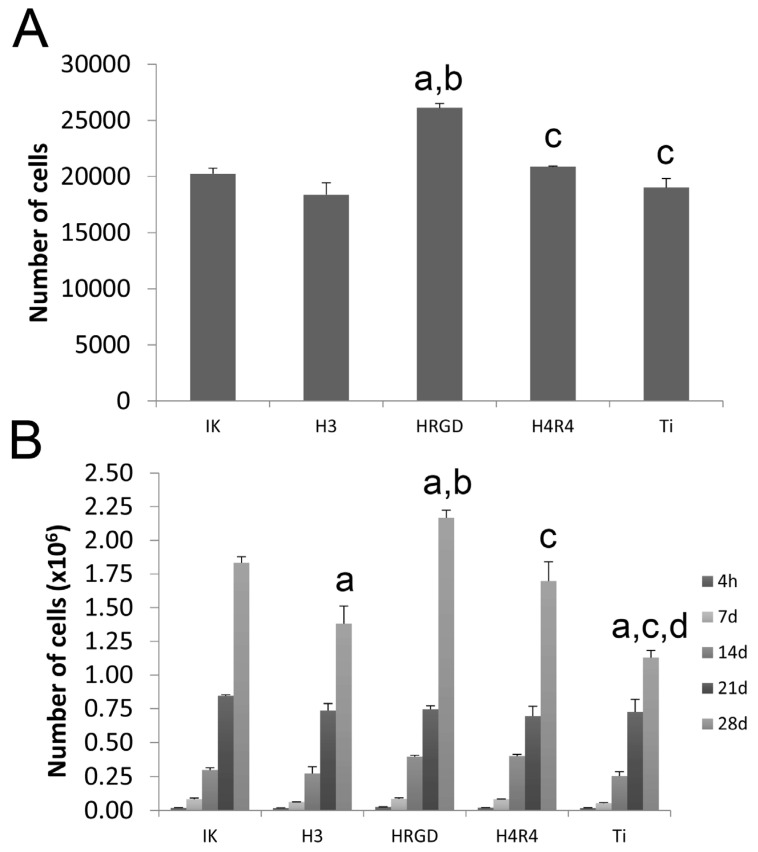
Cell adhesion at 4 h (**A**) and cell proliferation (**B**) of rMSCs on the different functionalized and non-functionalized (Ti) scaffolds. At each time point, “a” indicates statistically significant differences compared to IK, “b” indicates statistically significant differences compared to H3, “c” indicates statistically significant differences compared to HRGD, and “d” indicates statistically significant differences compared to H4R4.

**Figure 7 pharmaceutics-15-00872-f007:**
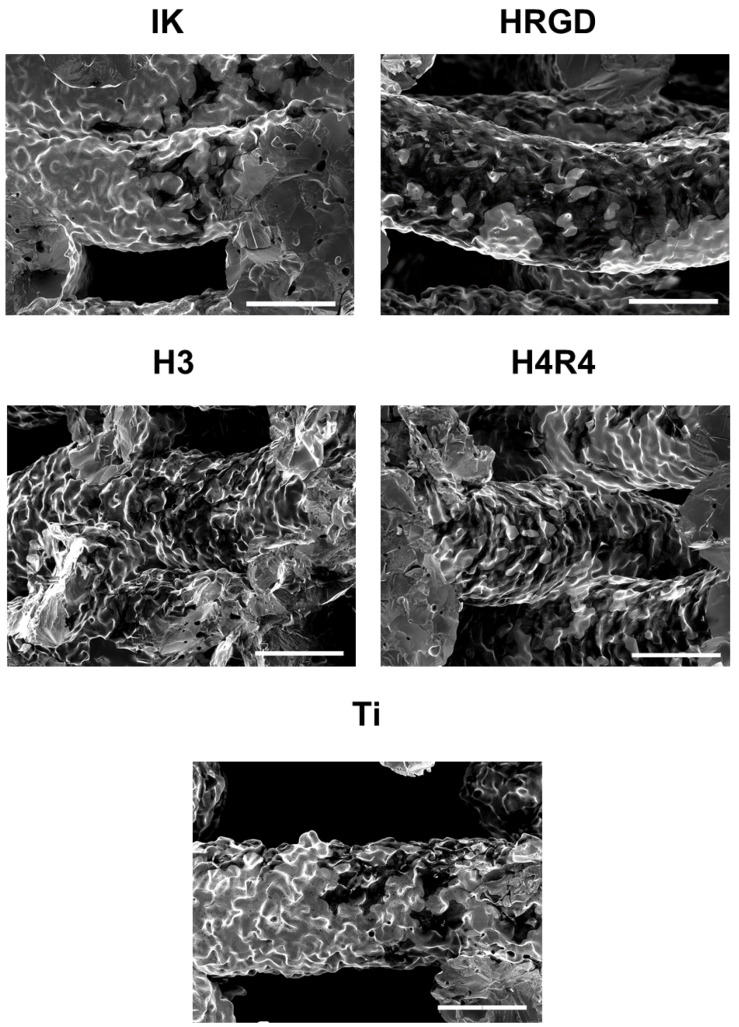
Representative SEM images of the different functionalized and non-functionalized Ti scaffolds showing attached cells 4 h after cell seeding. Scale bars denote 200 µm.

**Figure 8 pharmaceutics-15-00872-f008:**
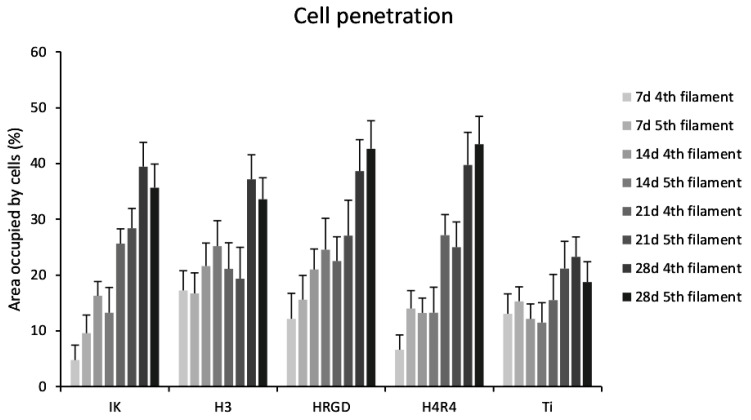
Quantification of the area occupied by cells on the parallel fourth and fifth filament (from the top downwards) at 7, 14, 21, and 28 days after rMSCs seeding on the different functionalized and non-functionalized Ti scaffolds.

**Figure 9 pharmaceutics-15-00872-f009:**
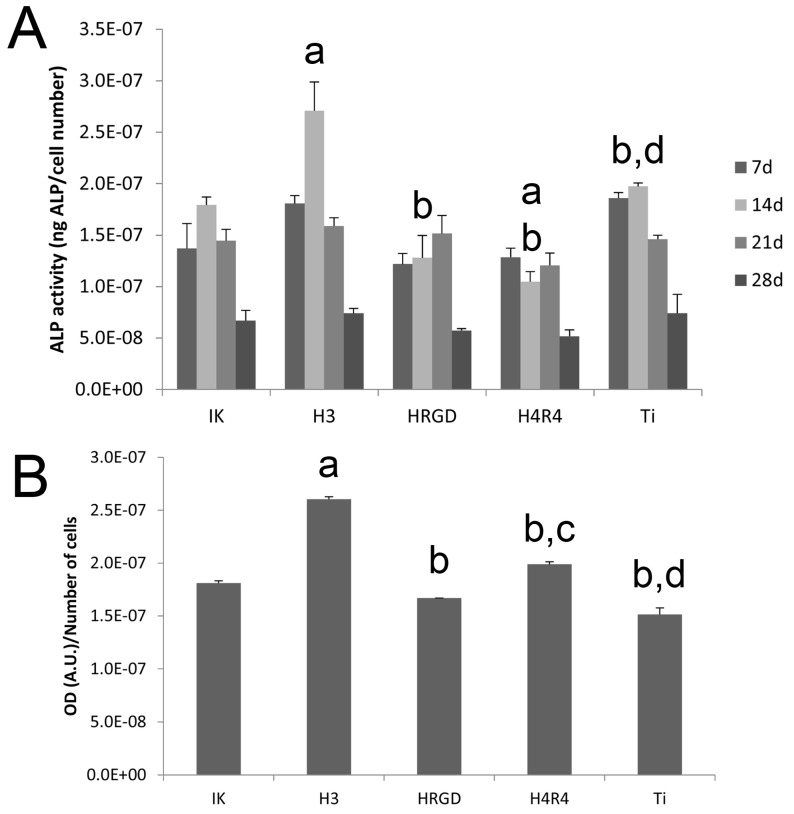
Cell differentiation: ALP activity at 7, 14, 21, and 28 days (**A**) and quantification of calcium deposits at 28 days (**B**) of rMSCs cultured on the different functionalized and non-functionalized (Ti) scaffolds. At each time point, “a” indicates statistically significant differences compared to IK, “b” indicates statistically significant differences compared to H3, “c” indicates statistically significant differences compared to HRGD, and “d” indicates statistically significant differences compared to H4R4.

**Table 1 pharmaceutics-15-00872-t001:** Amino acid sequences of the different ELRs. The hydroxyapatite nucleation and the adhesion sequences are highlighted in bold and underlined.

ELR	Amino Acid Sequence	MW Da
IK	MESLLP[(VPGIG)2(VPGKG)(VPGIG)2]_24_V	51,980
H3	MESLLP{[(VPGIG)_2_(VPGKG)(VPGIG)_2_]_2_ **DDDEEKFLRRIGRF**G [(VPGIG)_2_(VPGKG)(VPGIG)_2_]_2_}_3_V	31,877
HRGD	MGSSHHHHHHSSGLVPRGSHMESLLP{[(VPGIG)_2_(VPGKG)(VPGIG)_2_]_2_ AVTG**RGD**SPASS[(VPGIG)_2_(VPGKG)(VPGIG)_2_]_2_}_6_V	60,661
H4R4	MESLLP{[(VPGIG)_2_(VPGKG)(VPGIG)_2_]_2_ **DDDEEKFLRRIGRF**G [(VPGIG)_2_(VPGKG)(VPGIG)_2_]_2_]_4_[[(VPGIG)_2_(VPGKG)(VPGIG)_2_] _2_AVTG**RGD**SPASS[(VPGIG)_2_(VPGKG)(VPGIG)_2_]_2_]_4_}V	80,730

## Data Availability

The data presented in this study are available on request from the corresponding authors.

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
