# Peer review of "Functionalization of 3D-Printed Titanium Scaffolds with Elastin-like Recombinamers to Improve Cell Colonization and Osteoinduction"

_pharmaceutics, 2023, doi:10.3390/pharmaceutics15030872_

Round 1

Reviewer 1 Report

1.      The abstract should be broadened to give additional quantitative results.

2.      Please end your abstract with a "take-home" message.

3.      Reorder keywords based on alphabetical order.

4.      It is unclear whether the author's something new in this work. According to evaluation, several published studies by other researchers in the past adequately explain the issues you made in the present paper. Please be careful to highlight in the introduction section anything really innovative in this work.

5.      The work, novelty, and constraints of relevant previous research must be explained in the introduction section to highlight the research gaps that the present study aims to fill.

6.      Explain specifically the objective of the present study in the last paragraph of the introduction section.

7.      Line-41-42, the authors explain about the reasons for adopting titanium materials. The reasons of adopting titanium materials is also from aspect of corrosion resistance. The MDPI's suggested reverence should be adopted as follows: In Silico Contact Pressure of Metal-on-Metal Total Hip Implant with Different Materials Subjected to Gait Loading. Metals (Basel). 2022, 12, 1241. https://doi.org/10.3390/met12081241

8.      In order to improve the reader's understanding of the materials and methods section simpler, the authors could provide a figure that clarify the workflow of the current study rather than only the predominant text as it currently appears.

9.      More information about tools, such as the producer, country, and specifications, should be included.

Author Response

Please see the attacchment

Reviewer 2 Report

This manuscript studies cell colonization and osseointegration enhancement by synthesizing Elastin-like recombinamers (ELRs) into 3D-printed Titanium scaffolds. The topic holds significant importance to the field as proper functionalization of 3D-printed Ti scaffolds is one of the biggest hurdles in the synthetic bone implant sector. Yet, this reviewer thinks the manuscript has room for improvement by addressing the following comments.

-          The abstract starts strong but ends with a very weak tone and sounds passive. This could be fixed by including ‘quantified’ performance improvement of ELR-infused Ti scaffolds. If this is not doable, add a couple of sentences that showcase how the method could be improved further to show the potential of the work.

-          Overall manuscript feels unnecessarily dry. While the text content could be enough to make the experienced reader understand the material easily, adding figures and diagrams could enhance the understandability of readers without having profound knowledge. For instance, it would be much easier for readers to visualize things if proper drawings or photos were included in chapter 2 (materials and methods).

-          Most figures in the manuscript have poor quality (i.e., Figures 1, 3, 5, and 6). For instance, the plots in figure 1 (B and C), it is impossible to read axis labels even when zoomed in. This reviewer suggests providing vector graphics instead of raster images. Also, the fonts in the two graphs in figure 3 could be normalized to be a similar size to make them look professional.

-          Section 2.1 mentions the following

o   (line 107) 3D Ti scaffolds were designed using SolidWorks…: As the authors mentioned, the external shape and the internal pore architecture of 3D scaffolds affect osseointegration and cell colonization. Yet, these geometrical features of 3D Ti scaffolds are not discussed at all. There is a huge research field just on optimizing the scaffold internal geometries for different applications, so it would be better to elaborate on how it was designed and why the authors chose such a design; adding a photo would be good here.

o   (line 108) printing parameters were adjusted using the Slic3r software…: It is widely known that printing parameters affect the mechanical properties of the final printed structure. The precise printing parameters must be disclosed to improve the credibility of the work. 

Round 2

Reviewer 1 Report

Further comments is given in this stage as follows:

1.      Line 126, the authors mention porosity for the first time. Porosity is one of the urgent aspects of scaffold impacting their degradation performance. The authors needs to explain in brief the importance of porosity in the manuscript. Refer and discuss the relevant reference for this purpose as follows: Level of Activity Changes Increases the Fatigue Life of the Porous Magnesium Scaffold, as Observed in Dynamic Immersion Tests, over Time. Sustainability 2023, 15, 823. https://doi.org/10.3390/su15010823

2.      The revised manuscript after peer review must provide detailed information on the error and tolerance of the experimental equipment utilized in this study. Due to the disparate outcomes of other researchers' subsequent studies, it would make for a valuable discussion.

3.      Outcomes must be compared to similar past research.

4.      The authors need to improve the discussion in the present article become more comprehensive. The present form was insufficient.

5.      What is the current work's limitation? Please place it before entering the conclusion section.

6.      In the conclusion, please explain the further research.

7.      Five years back literature should be enriched into the reference. MDPI reference is strongly recommended.

8.      Because of grammatical faults and linguistic style, the authors must proofread the document.

9.      It is suggested to the authors for providing graphical abstract in the system after revision.

Round 3

Reviewer 1 Report

I have one minor comments to the authors in this stage.

The potential study adopting computational simulation via finite element analysis (in silico) for investigation of bone scaffold needs to be discussed to extend the explanation to become more comprehensive. The method offers faster results and lower cost compared to experimental analysis (in vitro) such as fester results and lower cost. To support this explanation, suggested reference should be included as follows: The Effect of Tortuosity on Permeability of Porous Scaffold. Biomedicines 2023, 11, 427. https://doi.org/10.3390/biomedicines11020427